# Anti-Hyperuricemic Effect of Anserine Based on the Gut–Kidney Axis: Integrated Analysis of Metagenomics and Metabolomics

**DOI:** 10.3390/nu15040969

**Published:** 2023-02-15

**Authors:** Mairepaiti Halimulati, Ruoyu Wang, Sumiya Aihemaitijiang, Xiaojie Huang, Chen Ye, Zongfeng Zhang, Lutong Li, Wenli Zhu, Zhaofeng Zhang, Lixia He

**Affiliations:** 1Department of Nutrition and Food Hygiene, School of Public Health, Peking University, Beijing 100871, China; 2Beijing’s Key Laboratory of Food Safety Toxicology Research and Evaluation, Beijing 100871, China; 3Division of Molecular and Cellular Oncology, Dana-Farber Cancer Institute, Brigham and Women’s Hospital, Harvard Medical School, Boston, MA 02115, USA

**Keywords:** hyperuricemia, anserine, metagenomics, urine metabolomics, integrated analysis, gut–kidney axis

## Abstract

Nowadays, developing effective intervention substances for hyperuricemia has become a public health issue. Herein, the therapeutic ability of anserine, a bioactive peptide, was validated through a comprehensive multiomics analysis of a rat model of hyperuricemia. Anserine was observed to improve liver and kidney function and modulate urate-related transporter expressions in the kidneys. Urine metabolomics showed that 15 and 9 metabolites were significantly increased and decreased, respectively, in hyperuricemic rats after the anserine intervention. Key metabolites such as fructose, xylose, methionine, erythronic acid, glucaric acid, pipecolic acid and trans-ferulic acid were associated with ameliorating kidney injury. Additionally, anserine regularly changed the gut microbiota, thereby ameliorating purine metabolism abnormalities and alleviating inflammatory responses. The integrated multiomics analysis indicated that *Saccharomyces*, *Parasutterella excrementihominis* and *Emergencia timonensis* were strongly associated with key differential metabolites. Therefore, we propose that anserine improved hyperuricemia via the gut–kidney axis, highlighting its potential in preventing and treating hyperuricemia.

## 1. Introduction

With advancements in economic growth and changes in lifestyles, the number of people suffering from hyperuricemia is increasing annually worldwide. Approximately 20.2% and 20.0% of American males and females, respectively, are estimated to suffer from hyperuricemia [1]. As the largest developing country, the incidence of hyperuricemia in China has been reported to be up to 17.4%, with the age of onset gradually decreasing [2]. Regardless of the presence of symptoms, hyperuricemia has tremendous harmful effects on the body, posing a challenge to patients and clinicians. In addition to being the main hazard for gout, hyperuricemia also increases the risk of other diseases [3], such as diabetes and chronic kidney disease. Furthermore, a recent Irish study illustrated that compared to the normal population, blood uric acid of males and females above 535 μM/L and 416 μM/L, respectively, decreased their median survival years by 11.7 and 6 years, respectively [4]. Therefore, there is an urgent need to identify effective measures to prevent and treat hyperuricemia.

As an end product of purine metabolism, uric acid plays an important antioxidative role similar to Vitamin C [5]. Hyperuricemia occurs as a result of purine metabolism dysregulation and has a complex pathogenesis. The risk factors of hyperuricemia include genetics, nutritional status, sleep and stress, which consequently induce inflammation, oxidative stress and insulin resistance in the body. Moreover, it is accompanied by liver and kidney damage and toxic epidermal necrolysis [6]. Under these circumstances, the body compensates by increasing uric acid levels to counteract the damage caused by the metabolic disorder.

Current treatments for hyperuricemia include xanthine oxidase inhibitors (allopurinol or febuxostat), which decrease uric acid production, and uricosurics (probenecid), which increase uric acid excretion. Additionally, a low-purine diet has been recommended for lowering blood uric acid levels. Although this diet is safe, the effects of dietary interventions on hyperuricemia are limited. A recent meta-analysis on the effects of diet and genetics on blood uric acid concentrations in Caucasians in New Zealand demonstrated that 63 types of food combinations accounted for only 4.29% of the variation in blood uric acid concentrations. Contrastingly, 23.8% and 40.3% variations in blood uric acid concentrations in males and females, respectively, were due to genetic factors [7]. Therefore, simply reducing uric acid concentrations does not deter the development of hyperuricemia. Furthermore, there is a need to elucidate the pathogenesis of hyperuricemia and identify an efficient dietary intervention using natural products to improve overall body metabolism.

Overproduced uric acid or reduced uric acid excretion are the two primary causes of hyperuricemia. The kidneys excrete two-thirds of the uric acid, while the intestines excrete a third [8]. Excretory disorders related to the kidney are usually associated with the regulation of molecular signals, such as insulin resistance, inflammation, oxidative stress and cell damage [5]. Low-grade chronic systemic inflammation can directly lead to kidney damage and then affect the kidney’s uric acid-related transporters expression, ultimately affecting the excretion of uric acid [9]. Additionally, studies show that high uric acid impairs mitochondrial function and produces reactive oxygen species, which activates the inflammatory bodies of the NOD-like receptor (NLR) family containing pyrimidine domain 3 (NLRP3). This cascade further aggravates kidney injury by secreting interleukin-1β (IL-1β) [10]. Moreover, excess uric acid enters the cells and becomes pro-oxidative, subsequently causing oxidative stress, aggravating kidney dysfunction and improving insulin resistance, thus forming a vicious circle. Therefore, reducing oxidative stress and low-grade chronic inflammation in the kidney is vital for the excretion of uric acid via the kidney. Various metabolites produced during this pathophysiological process are ultimately excreted by the kidneys through urine. Therefore, exploring urinary metabolites in hyperuricemia could provide a more comprehensive elucidation of its pathogenesis.

Recently, the gut microbiota has been reported to influence hyperuricemia, especially via the gut–kidney axis [11]. Then, the kidneys are damaged due to hyperuricemia, and uric acid and uremic toxins accumulate in the blood [12]. Moreover, low-grade inflammation produces proinflammatory cytokines. Elevated uric acid levels, uremic toxins and inflammatory cytokines negatively affect gut microbiota homeostasis. Dysbiosis of the gut microbiota increases intestinal permeability, which allows bacteria and intestinal metabolites, such as lipopolysaccharide (LPS), to be transported out of the intestine. LPS forms a complex with its CD14 receptor and is detected by Toll-like receptor 4 (TLR4), which induces chronic low-grade inflammation and thereby aggravates kidney injury [13]. The entire process forms a vicious cycle and increases the risk of hyperuricemia. Furthermore, gut dysbacteriosis increases the abundance of the xanthine oxidase gene-related microbiota and decreases the abundance of allantoinase gene-related flora, resulting in elevated uric acid levels in the intestinal tract. Therefore, this study aims to explore intervening substances that act on the gut–kidney axis.

The ocean is a uniquely rich source of bioactive peptides. In studies on peptides associated with hyperuricemia, pelagic fishes, such as tuna and bonito, have been found to migrate tens of thousands of kilometres at high speeds without causing an acid build-up in the body. This phenomenon was attributed to the presence of an important dipeptide, anserine, in the body, which has currently become a new target for dietary intervention in hyperuricemia [14]. Anserine is a multifunctional and highly stable histidine carnosine-like dipeptide found in fish skeletal muscles. In human clinical trials, anserine has been shown to reduce blood glucose and inflammation and elevate kidney functions [15]. Additionally, it improved gut microbiota disorders caused by hyperuricemia in mice [16]. However, the effect of anserine on host metabolism remains unclear and the alleviation of hyperuricemia via the gut–kidney axis remains to be investigated.

Thus, this study establishes a rat model for hyperuricemia using potassium oxycyanate and yeast and verifies the ameliorating effect of anserine on hyperuricemia. It also explores the mechanism of anserine-related amelioration of hyperuricemia through gut microbiota and metabolites using a combined ultraperformance liquid chromatography–tandem mass spectrometry (UPLC–MS) and macrogenomic analysis. Thus, this study aims to explore novel ideas for preventing and treating hyperuricemia.

## 2. Materials and Methods

### 2.1. Materials and Reagents

A rodent regular diet was obtained from Beijing Weitong Lihua Laboratory Animal Technology Co., Housed in a clean-grade animal house at the Peking University Health Science Center and the provided nutrients met the needs for rodent growth and development per GB 14924.3-2010. The diet for hyperuricemic rats comprised a normal diet combined with 4% potassium oxonate and 20% yeast, which was obtained from Beijing BotaiHongda Biotechnology Co., Ltd. Anserine was obtained from Sinopharm Holding Starshark Pharmaceutical (Xiamen) Co., Ltd. Allopurinol was obtained from Beijing Balinwei Technology Co., Ltd.

Primary antibodies against TLR4, myeloid differentiation factor88 (MyD88), nuclear factor E2-related factor 2 (Nrf2) antibody, NLRP3 inflammation, ATP-binding cassette, subfamily G, member 2 (ABCG2), matrix metallopeptidase 2 (MMP2), matrix metallopeptidase 2 (MMP9), β-actin and secondary antibodies were obtained from Abcam Co., Ltd. The primary antibody nuclear factor-κB (NF-kB) was obtained from CST Co., Ltd., whereas that against urate transporter 1 (URAT1) was obtained from Proteintech Biotech Co., Ltd. The primary antibodies against glucose transporter 9 (GLUT9) and the tissue inhibitor of metalloproteinase1 (TIMP-1) were obtained from Novus Biologicals Co., Ltd. and BOSTER Co., Ltd., respectively.

### 2.2. Animal Treatment

A total of 60 male Sprague–Dawley rats (180–220 g) were obtained from the Beijing Vital River Laboratory Animal Technology Co., Ltd. The animals were housed in a climate-controlled room at 25 ± 1 °C with a 12/12 h dark/light cycle and free access to sterile water and standard feed. All laboratory procedures were reviewed and approved by the Institutional Animal Care and Use Committee of Peking University (Approved No. 2019PHE017) on the 5th May 2019 and conformed to the Guidelines for the Care and Use of Laboratory Animals (NIH Publication No. 85-23, Revised 1996).

After four days of acclimatization, the rats were randomly divided into six groups (n = 10): normal control group (NC group), hyperuricemia group (HUA group), allopurinol group (Allo group, 10 mg/kg·bw allopurinol), three anserine groups (Ans1, Ans10 and Ans100 groups were treated with 1 mg/kg·bw, 10 mg/kg·bw and 100 mg/kg·bw anserine, respectively). During the six weeks of intervention, the general conditions of experimental animals, including health and behaviour, were observed daily, and food intake, water intake, urine volume and weight gain were monitored weekly.

### 2.3. Plasma and Urine Biochemical Analysis

Urine was collected for 24 h after six weeks of intervention and the volume was measured. Following centrifugation at 3000 r/min for 10 min, the supernatant was collected and stored at −80 °C. Subsequently, blood samples were collected and centrifuged at 3000 r/min for 10 min at 4 °C after the rats were sacrificed. The concentrations of serum uric acid (SUA), serum creatinine (SCr), blood urea nitrogen (BUN), urinary uric acid (UUA), urinary creatinine (UCr), alanine aminotransferase (ALT), aspartate aminotransferase (AST), albumin (ALB), globulin (GLB), albumin/globulin (A/G) and total protein (TP) were measured using an automatic biochemical instrument (AU480, Japan Olympus Corporation). Uric acid clearance (CUA) and creatinine clearance (CCr) were calculated using the following formulas:CUA = UUA/SUA × Urine volume (mL/min); CCr = UCr/SCr × Urine volume (mL/min).

### 2.4. Serum and Liver Xanthine Oxidase (XOD) and Adenosine Deaminase (ADA) Assay

Serum XOD activity was measured using the colourimetric method. After the liver was excised, 1 g of liver tissue was added to 9 mL of 0.9% normal saline to prepare 10% liver homogenate using mechanical homogenizers. The homogenate was centrifuged at 3000 r/min for 10 min and then 100 μL of the 10% liver homogenate supernatant was collected to evaluate liver XOD activity using colourimetry. A total of 50 μL of the liver homogenate supernatant was used to determine ADA activity. ADA generates ammonia by hydrolysing adenosine, which is then used to calculate ADA activity.

### 2.5. Histopathological Evaluation

The rats were sacrificed after the experiment. Then, their right kidneys were fixed with 4% paraformaldehyde, and paraffin sections were made by a series of techniques such as dewaxing, hematoxylin staining, eosin staining, dehydration and sealing. Finally, the tissue was examined under an optical microscope (Olympus BX43) and the image was recorded. The remaining half of the right kidney was fixed with 4% glutaraldehyde at a size of 1 mm × 1 mm × 1 mm, and then underwent a series of processes such as dehydration, transparency, wax immersion, embedding and sectioning to obtain electron microscopic sections for observation under an electron microscope (JEM1400).

### 2.6. Western Blotting

The expressions of GLUT9, URAT1, ABCG2, TLR4, MyD88, NF-κB, NLRP3, Nrf2, MMP-1, MMP-2 and TIMP-1 proteins in the renal tissue were detected using Western blotting. Commercial kits (Beijing Prily Gene Technology Co., Ltd. C1053) were used to extract total protein from kidney tissue, and BCA protein detection kits (Beijing Prily Gene Technology Co., Ltd. P1511) were used to quantify protein concentrations. After protein separation using 12% sodium dodecyl-sulphate polyacrylamide gel electrophoresis, an Immobilon-P transfer membrane (Millipore, Germany, IPVH00010) was used for transferring the protein. Following this, the membrane was sealed with 5% nonfat milk, incubated overnight at 4 °C with primary antibodies, and then incubated with secondary antibodies conjugated with horseradish peroxidase. Colour development using ECL Chemiluminescence Chromogenic solution (PerkinElmer, America, NEL105001EA) was performed. The grayscale was calculated and normalised to the values of β-actin.

### 2.7. Untargeted Urine Metabolomic Analysis

To 100 μL of urine sample in an Eppendorf tube, 400 μL of 80% methanol aqueous solution (Thermo Fisher, Waltham, MA, USA, 67-56-1) was added. The sample was vortexed, placed in an ice bath for 5 min and centrifuged at 15,000 r/min for 20 min at 4 °C. A quantity of the supernatant was diluted with MS-grade water to obtain a methanol content of 53%. This sample was then centrifuged at 15,000 r/min for 20 min at 4 °C and the supernatant was analysed using a UPLC–MS. Metabolomics workflow was based on the R language MetaboAnalystR package. First, the sample data were subjected to quality control and batch correction, then the sample data were standardized, and finally, the metabolite content was counted. Metabolites that differed between groups were identified using an orthogonal partial least squares discriminant method.

### 2.8. Metagenomic Sequencing of Gut Microbiota

DNA was extracted from stool samples using the CTAB method. After the DNA samples were qualified, the library was constructed. Qubit 2.0 was used for preliminary quantification, and then, an Agilent 2100 was used for the detection of inserted fragments from the library. Using real-time polymerase chain reaction, the effective concentration of the library could be accurately quantified after the insert size reached the expected size. Following library detection, different libraries were pooled into a flow-through pool based on effective concentration and target data volume. After cBOT clustering, the Illumina PE150 (2 × 150) high-throughput sequencing platform was used for sequencing. KneadData software was used for raw data quality control (Trimmom-based) and de-hosting (Bowtie2-2-based). Before and after KneadData, FastQC was used to verify the justification and effectiveness of the quality control. Furthermore, Kraken2 was used to analyse the species composition and diversity information of samples, whereas Bracken was used for predicting species relative abundances. The identification of microbial genera and species that differed between the six groups was performed using LEfSe biomarker mining analysis (microorganisms with LDA > 2 by default). To functionally annotate genes, we used HUMAnN2 software to compare the sequences after quality control (based on DIAMOND) and host removal with the protein database (UniRef90), further filtering out reads that failed to be compared. Based on the mapping between the ID of UniRef90 and the ID of the KEGG and EC functional databases, the relative abundance of the corresponding functions of each functional database was calculated.

### 2.9. Statistical Analysis

#### 2.9.1. Statistical Analysis of Non-Omics Data

One-way analysis of variance (ANOVA) was calculated using SPSS 25.0 software; variables were transformed if they did not meet the requirements of normality for variance. If the transformed variables still did not meet the requirements, a nonparametric test was used for statistical analysis. The experimental and control groups were compared using the least significant difference (LSD) method, and a difference of *p* < 0.05 was considered to be statistically significant.

#### 2.9.2. Statistical Analysis of Omics Data

The omics data were analysed using R software (version 4.1.1). Principal coordinates analysis (PcoA) and principal components analysis (PCA) were used for dimension reduction analysis to show the degree of variation among the samples. Binary unpaired samples were tested for significance using the rank-sum test, and multiple groups were compared using the Kruskal–Wallis rank-sum test.

#### 2.9.3. Multiomics Association Analysis

The correlation between gut microbiota and metabolites was analysed using the Pearson correlation coefficient method, with a significance threshold of *p* < 0.05.

## 3. Results

### 3.1. Anserine Reduced Uric Acid Levels and Improved Kidney Damage

The HUA group had a significantly higher SUA than the NC group after 6 weeks of potassium oxalate and yeast feeding (*p* < 0.05), indicating successful modelling. After the anserine intervention, the SUA level was significantly decreased (*p* < 0.05). Moreover, BUN was significantly elevated in the HUA group but tended to decrease after the anserine intervention. SCr level was not significantly elevated in the HUA group; however, it was notably decreased on the high-dose anserine intervention compared to the HUA group (Figure 1A–C).

Hyperuricemia is often related to impaired kidney function, which can be reflected by CUA and CCr levels. CUA and CCr levels were significantly decreased in the HUA group but elevated on anserine intervention. Notably, the levels of CCr were elevated in both the HUA group and the anserine intervention groups; however, the anserine groups exhibited higher levels. In the Allo group, there were no significant differences in SUA, BUN or SCr levels compared with the HUA group; however, CUA and CCr levels were elevated (Figure 1D,E).

### 3.2. Anserine Inhibited Uric Acid Production and Improved Liver Function

Serum XOD and ADA in the HUA group were higher than in the NC group; however, serum ADA was decreased in the Ans100 group and Allo group (*p* < 0.05) (Appendix A). Additionally, liver XOD levels did not significantly differ among the groups (Appendix A). In the HUA group, serum ALT and AST levels were increased and A/G levels were decreased compared with the NC group; however, anserine intervention altered these changes and showed similar levels to that of the NC group. Furthermore, allopurinol intervention showed similar effects as those of anserine (*p* < 0.05) (Appendix A).

### 3.3. Anserine Promoted Uric Acid Excretion by Regulating Kidney Urate Transport-Related Proteins

We further investigated the excretion-related proteins of the kidney. URAT1 and GLUT9 mainly mediate uric acid reabsorption, while ABCG2 primarily mediates uric acid excretion. In this study, URAT1 and GLUT9 expressions showed no significant difference in the NC and HUA groups; however, they were notably decreased by anserine intervention in the Ans group (*p* < 0.05). Moreover, ABCG2 expression in the HUA group was lower than that in the NC group but was subsequently reversed by anserine, (*p* < 0.05). However, there was no significant difference between the Allo and HUA groups (Figure 2A–C).

### 3.4. Anserine Ameliorated Pathological Changes in the Kidneys in Hyperuricemia Rats

To investigate kidney damage in hyperuricemic rats, a histological evaluation was performed (Figure 3). The kidney surface of the HUA rats was rough and vacuolated; however, after the anserine intervention, the surface became smooth and the vacuole disappeared. Furthermore, in HUA rats, the tubular lumen was dilated and inflammatory cell infiltration was observed in the kidneys, which was greatly improved on anserine intervention. Electron microscopy further showed that the epithelial cells were swollen and kidney podocyte fusion was observed in the HUA group; however, these changes were also reversed on anserine intervention. Although the kidney surface roughness was improved in the Allo group compared to the HUA group, the phenomenon of enlargement and vacuolation along with podocyte fusion remained.

### 3.5. Effects of Anserine on the Kidney TLR4/MyD88/NF-κB Pathway, NLRP3 Inflammasome, Nrf2 and Cell Damage-Related Proteins

In our study, compared to the NC group, the expressions of TLR4, MyD88 and NF-κBp65 were enhanced slightly in the HUA group but were decreased on anserine intervention (*p* < 0.05) (Figure 4A–C). Compared to the NC group, NLRP3 expression was increased in the HUA group but decreased on the anserine and allopurinol interventions (*p* < 0.05) (Figure 4D). Moreover, the Nrf2 expression in the NC and HUA groups did not differ significantly; however, the anserine intervention enhanced the Nrf2 expression, while the allopurinol intervention reduced it (*p* < 0.05) (Figure 4E). Furthermore, neither the MMP-2 nor the MMP-9 expression significantly differed among the groups (Figure 4F–H). However, compared to the NC group, the TIMP-1 level in the HUA group decreased but was reversed on the anserine intervention (*p* < 0.05).

### 3.6. Metagenomic Sequencing of Gut Microbiota

#### 3.6.1. Overview of the Changes in Gut Metagenomic Diversity

Significant differences were observed between the gut microbiota structures of the HUA and NC groups and between the anserine dose and HUA groups based on the PCA and PcoA results. (Figure 5A,B) (*p* < 0.05). Intestinal flora diversity was analysed using the Shannon index. The HUA group had lower diversity than the NC group, which was reversed by anserine treatment. However, allopurinol failed to elicit a significant effect (Figure 5C) (*p* < 0.05).

#### 3.6.2. Changes in Gut Microbiota at Different Levels

At the phylum level, compared to the NC group, the abundance of *Bacteroidetes* and *Proteobacteria* was increased and that of *Firmicutes* was decreased in the HUA group; however, anserine and allopurinol treatment reversed these changes (Figure 6A) (*p* < 0.05). Seven different families were identified by LDA analysis. At the family level, the abundance of *Lactobacillaceae* in the HUA group was the lowest, while that of *Lachnospiraceae* and *Clostridiales* was the highest (Figure 6B) (*p* < 0.05).

Furthermore, the lesser method was used to obtain 79 different genera between the five comparison groups (NC vs. HUA, HUA vs. Ans100, HUA vs. Ans 10, HUA vs. Ans1, HUA vs. Allo) (Appendix A), with *Saccharomyces* being the common differential genus (Figure 6C). Heatmap analysis revealed the relative abundances of these 79 differential genera among the six groups (Figure 6D). *Alcaligenes*, *Emergencia* and *Lachnoclostridium* were the highest while *Saccharomyces*, *Roseburia* and *Coprococcus* were the lowest in the HUA group (Figure 6E) (*p* < 0.05).

At the species level, based on the results of the OUT comparison, a total of 2568 microbial species were identified. The lesser analysis yielded a total of 189 differential species between the five comparison groups (Appendix A), with *Saccharomyces cerevisiae* being the common differential species (Figure 6F) (*p* < 0.05).

#### 3.6.3. Changes in the Metabolic Function of Gut Microbiota

Furthermore, the intestinal flora-related function changes after anserine intervention were examined. Based on the KEGG database, at level 1, the relative abundance of metabolic pathways was the highest in each group, followed by cellular processes (Appendix A). Furthermore, KEGG included 42 subdivided pathways, and differential analysis yielded 25 differential pathways among the groups (Appendix A). The comparison of the abundance of the KEGG map and KO enrichment revealed a heatmap that showed the relative abundance of the top 20 differential maps and the top 30 differential KOs among the six groups (Appendix A). Among them, three metabolic pathways in the HUA group were enriched compared to other groups, including D-Arginine and D-ornithine metabolism and Lipoarabinomannan (LAM) biosynthesis (Figure 7A). An in-depth analysis of the D-Arginine and D-ornithine metabolism pathways revealed that the genes 2,4-diaminopentanoate dehydrogenase and D-ornithine 4,5-aminomutase subunit beta corresponding to two proteins ord and oraE were more abundant in the HUA group than other groups but were significantly reduced on anserine and allopurinol interventions (Figure 7B).

### 3.7. Urine Metabolomic Analysis

The HUA group was compared with five other groups to form five comparison groups. There were 38 differential metabolites in these five comparison groups (Figure 8A). Figure 5B demonstrates the changing trends in differential metabolites for the five comparison groups. Compared to the NC group, the HUA group had 21 different metabolites, with 11 showing a significant decrease, including galactose, mannose, threonine and D-glucuronic acid. However, inositol, uracil, γ-amino, valine and another 10 metabolites increased significantly in the HUA group. Compared to the HUA group, 15 metabolites were significantly increased and 9 metabolites were significantly decreased in the anserine intervention groups, with fructose, xylose, threonine D-glucuronide and methionine showing an increasing trend, and glycine and pipecolic acid showing a decreasing trend (Figure 8B). Moreover, four metabolites, namely erythronic acid, glucaric acid, pipecolic acid and trans-ferulic acid, were the common differential metabolites of the five comparison groups. Compared with the other groups, the contents of erythronic acid, glucaric acid and trans-ferulic acid in the HUA group were lower, while that of pipecolic acid was higher (Figure 8C).

### 3.8. Association Analysis of Differential Gut Microbiota and Differential Urinary Metabolites

We explored the correlation of 38 differential metabolites with 79 differential genera. The results showed that methionine was significantly correlated with 39 genera, which was the highest association (Appendix A). Among all genera, *Parasutterella*, *Oligella*, *Catabacter*, *Emergencia* and *Bacteroides* were significantly associated with 18, 15, 13, 13 and 13 metabolites, respectively, and formed the top five genera associated with most differential metabolites. Among them, *Saccharomyces* was only related to methionine (Appendix A). At the species level, we analysed 189 differences in species and approximately 38 differences in metabolites (Appendix A). Different species that were significantly associated with erythronic acid, glucaric acid, pipecolic acid and trans-ferulic acid were extracted, revealing three species, *Parasutterella excrementihominis*, *Emergencia timonensis* and *Bacteroides uniformis*. Considering that pipecolic acid and differential species *Parasutterella excrementihominis* and *Emergencia timonensis* were more abundant in the HUA group than in the other five groups, we speculated that there was a positive correlation between them. Conversely, erythronic acid, glucaric acid and trans-ferulic acid were negatively correlated with the abundance of *Parasutterella excrementihominis* and *Emergencia timonensis* (Figure 9). Furthermore, glucaric acid was associated with the greatest number of differential species (Appendix A).

## 4. Discussion

This study used yeast combined with potassium oxalate to establish a rat model with hyperuricemia. Anserine was found to significantly reduce blood uric acid levels and improve liver and kidney damage, indicating that anserine had a preventative effect on hyperuricemia. Additionally, the changes in the gut microbiota structure and function as well as host metabolism, which were induced by hyperuricemia, were partially reversed by anserine. Moreover, the underlying mechanism of anserine ameliorating hyperuricemia was explored using integrated macrogenomics and metabolomics for the first time, to the best of our knowledge, in this study.

It is reported that, in general, bonito, bigeye tuna, and southern tuna contain about 1070 mg, 1260 mg and 636 mg of anserine per 100 g of fish flesh, respectively [17]. We learned that the doses of anserine used in previous animal experiments ranged from 2 mg/kg·bw to 80 mg/kg·bw [18,19], so we chose the 1 mg/10 mg/100 mg kg·bw as the low/medium/high doses of anserine intervention in our experiment. Our results did show that medium doses of anserine were the most effective in improving hyperuricemia. Consequently, we speculate that the effect of anserine in improving hyperuricemia is dose-dependent. Therefore, in subsequent experiments using Western blotting to detect the expression of uric acid-related transporter proteins and proteins associated with renal inflammation, oxidative stress, and cellular damage in the kidney, we only examined kidney samples from the mid-dose group of the goose-peptide intervention.

Initially, the effect of anserine on uric acid production and excretion was investigated. Anserine showed no obvious effects on ADA and XOD enzymes, which are closely related to uric acid production. However, anserine ameliorated liver damage induced by hyperuricemia. Then, the effect of anserine on kidney uric acid excretion was investigated. URAT1, GLUT9 and ABCG2 expression changes indicated that anserine reduced serum uric acid levels by inhibiting uric acid reabsorption and promoting uric acid excretion. The changes in CUA and CCr, two indicators of kidney function, also implied that anserine improved kidney function, which is consistent with the morphological changes observed in the kidneys. It is noteworthy that, in our experiments, the levels of CCr were elevated in the HUA group. We assume this is very likely because when the blood creatinine in hyperuricemic rats started to rise, the body compensated by increasing the renal creatinine clearance to keep the body’s creatinine level at a steady state. Additionally, the timing of the measurement also has some influence on the results. In short, this phenomenon is very interesting and deserves further in-depth study.

We further investigated the effect of anserine on kidney injury caused by hyperuricemia by evaluating the effect of anserine on kidney inflammation, oxidative stress and cellular injuries. The TLR4/MyD88/NF-κB and NLRP3 proteins are reported to be the major signalling pathways closely associated with renal inflammation caused by hyperuricemia [20]. Contrarily, Nrf-2 neutralises the activation of cellular oxidative stress and ameliorates kidney injury by inhibiting NF-κB expression [21]. Moreover, MMP2 and MMP9 are reported to cleave collagen IV in the basement membrane of cell bands and have activity in kidney tissue [22]. Furthermore, TIMP-1 inhibits the activities of most MMPs, thus improving cell damage [23]. Given that anserine decreased the expression of TLR4/MyD88/NF-κB and NLRP3 and elevated Nrf2 and TIMP1 in hyperuricemic rats, we speculate that anserine improves overall kidney function by decreasing inflammation, oxidative stress and cellular damage in hyperuricemia. Anserine also regulates the expressions of uric acid-related transporters, thereby reducing serum uric acid levels.

Increasing evidence suggests that gut microbiota balance is closely related to metabolic disorders, and patients with hyperuricemia have a different intestinal microbiota structure from normal individuals [24]. Abnormally high levels of uric acid in the blood enter the intestine and affect the steady state of the intestinal flora, thereby affecting the intestinal metabolism of uric acid and consequently aggravating hyperuricemia. Therefore, we analysed the effect of anserine on the changes in intestinal flora structure caused by hyperuricemia. Our study showed a decrease in the intestinal microbial diversity of hyperuricemic rats, which is consistent with previous studies [24]; however, this was reversed on anserine intervention.

Uric acid is the end product of purine metabolism; moreover, intestinal flora has also been shown to play an essential role in purine oxidative metabolism. For example, *Escherichia coli* in the human gut produces XOD to influence the production of uric acid [25]. The *Lactobacillaceae* family inhibits the growth of *E. coli* by secreting reuterin [26], indirectly inhibiting uric acid accumulation. Additionally, *Lactobacillus* can synthesize various UA metabolic enzymes, such as uricase, allantoinase and allantoicase, which can decompose uric acid into 5-hydroxyisothreonate, allantoin, allantoate and finally degrade it into urea [27]. Similarly, the *Clostridiaceae* family also degrades uric acid [28]. *Saccharomyces cerevisiae* is a fungus that secretes urate oxidase, which can catalyse uric acid oxidation and plays an essential role in the purine degradation pathway, thereby preventing uric acid accumulation [29]. This study showed that the abundance of *Lactobacillaceae*, *Clostridiaceae* family and *Saccharomyces cerevisiae* was reduced in hyperuricemic rats but elevated after the anserine intervention, suggesting the preventive effect of anserine was partially due to the changes in some specific microbiota.

A dysregulated gut microbiota is accompanied by imbalanced intestinal metabolites, such as trimethylamine, short-chain fatty acids (SCFAs) and LPS, which are considered mediators between the intestinal microbiome and their human hosts [30]. Anserine increases the abundance of *Roseburia* and *Coprococcus* in hyperuricemic rats, which are crucial in SCFA generation. SCFAs regulate gut microbiota homeostasis, repair intestinal permeability and are beneficial to kidney function [31]. Moreover, butyrate, a major SCFA in the intestine, is reported to be increased in a *Lactobacillaceae-*enriched environment [32]. Therefore, *Lactobacillaceae* is speculated to not only participate in purine metabolism but also play a role in increasing butyrate levels in the intestinal tract. Additionally, gut microbiome dysbiosis can cause the excretion of LPS from the cell walls of Gram-negative bacteria, and the inflammation in the liver and kidney is further activated by the excreted LPS entering the bloodstream through a disrupted gut barrier [33]. Furthermore, members of the Gram-negative *Proteobacteria phylum*, *Alcaligenes* genus and *Lachnoclostridium* genus were increased in the HUA group but were reduced on anserine intervention. Additionally, a Proteobacterial strain has also been shown to enhance intestinal nitrogen fixation [34], wherein, nitrogen is converted to ammonia. Notably, excess ammonia entering the host’s circulatory system through the intestinal barrier can aggravate kidney damage. Additionally, we observed the *Emergencia timonensis* genus was more enriched in the HUA group than in other groups. Furthermore, *Emergencia timonensis*, a potential key bacterium for the conversion of carnitine to trimethylamine N-oxide (TMAO), is also a toxin that can aggravate kidney damage [35]. This study indicated that anserine alleviated hyperuricemia owing to its ability to maintain the balance in the composition of the intestinal microbiota (the increase in beneficial bacteria and the decrease in pathogenic bacteria). Moreover, it also promotes purines and uric acid catabolism, regulates intestinal epithelial cell proliferation, reduces chronic inflammation and improves uric acid excretion.

The intestinal microflora greatly affects the health of the host by regulating its metabolic function. In this study, six metabolic pathways were altered in the HUA group compared to the NC group, three of which were elevated and the other three were decreased; however, anserine intervention reversed these changes. For example, D-Arginine and D-ornithine metabolism pathways were significantly enriched by hyperuricemia but anserine supplementation reversed the change, which was verified by the changes in two key proteins in this pathway, namely 4-diaminopentanoate dehydrogenase and D-ornithine 4,5-aminomutase subunit beta. The D-Arginine and D-ornithine metabolism are related to the urea cycle, indicating that more urea is metabolized in the intestine to produce ammonia in hyperuricemia. The disturbance of the gut microbiota combined with the reduction in beneficial metabolites such as SCFAs increases the permeability of the intestine, which increases the ammonia levels entering the circulation system, thereby aggravating liver and kidney function.

Metabolic profiling of host biofluids provides profound insights into the gut microbiota’s impact on host health/disease status, therefore exploring differential urinary metabolites aids in identifying the causative agent rather than the presence of the metabolite [36]. By comparing urinary metabolites in the HUA group with the different doses of the anserine group, Allo and NC groups, we identified erythronic acid, glucaric acid, pipecolic acid and trans-ferulic acid as the four common differential metabolites. This suggests that these four metabolites and their associated metabolic pathways play a critical role in the pathogenesis and amelioration of hyperuricemia. Erythronic acid is related to mitochondrial dysfunction in transaldolase deficiency [37], highlighting its role in mediating energy metabolism in humans. D-gluconic acid has toxin-reducing and antioxidant abilities, wherein it can improve diabetic kidney tubular damage by inhibiting inositol oxygenase, preventing mitochondrial damage and apoptosis and reducing oxidative stress through the ascorbic acid and aldehyde metabolic pathway, thereby improving kidney function [38]. Moreover, ferulic acid has been shown to lighten oxidative stress through the activation of the AMPK signalling pathway in vitro [39]. Pipecolic acid is an intermediate in the lysine degradation pathway, with an enhanced lysine degradation pathway indicating enhanced levels of oxidative stress in the host [40]. Thus, by enhancing the levels of erythronic acid, glucaric acid and trans−ferulic acid and decreasing the levels of pipecolic acid in hyperuricemic rats, anserine exerts an anti-hyperuricemia effect by improving energy metabolism and reducing oxidative stress and inflammation. Notably, *Parasutterella excrementihominis*, *Emergencia timonensis* and *Bacteroides uniformis* were associated with these four metabolites. As *Parasutterella excrementihominis* and *Emergencia timonensis* are positively associated with pipecolic acid but negatively associated with erythronic acid, glucaric acid and trans-ferulic acid, we speculated that anserine primarily reduced the abundance of *Parasutterella excrementihominis* and *Emergencia timonensis* to exert an ameliorating effect on kidney injuries.

Notably, methionine was associated with the highest number of differential genera and species; however, the *Saccharomyces* genus was only correlated with methionine. Methionine produces strong antioxidative metabolites such as glutathione, cysteine and sulphate through the trans-sulphuration pathway. Previous studies have demonstrated the ameliorative effect of methyl and S-adenosylmethionine produced by the methionine cycle on systemic inflammation and liver damage [41]. The methionine cycle is widely active in *Saccharomyces cerevisiae* [42]. This suggests that *Saccharomyces cerevisiae* could be a target probiotic for anserine to improve hyperuricemia. Additionally, the anserine group exhibited significantly higher levels of two differential metabolites (fructose and xylose) in the starch and sucrose metabolism pathway than the HUA group. Starch and sucrose metabolic pathways are directly related to the development of diseases involved in energy metabolism and insulin resistance, such as obesity, diabetes and kidney tubular dysfunction [43].

## 5. Conclusions

This study reveals the beneficial effects of anserine on the reversal of hyperuricemia. It exerts a beneficial effect by regulating intestinal microbiota and host metabolites. Additionally, the key differential metabolites that improve anserine-associated hyperuricemia were identified to be fructose, xylose, methionine, erythronic acid, glucaric acid, pipecolic acid and trans-ferulic acid, and key differential gut microbiota were identified to be *Saccharomyces*, *Parasutterella excrementihominis* and *Emergencia timonensis*, which are involved in the gut–kidney axis. These key microbiota and metabolites have the potential as disease markers to predict the onset of disease, thereby improving the efficiency and accuracy of early clinical diagnosis. However, this study also has some limitations. This study lacked gut metabolite and enzyme data related to uric acid metabolism, which requires further study. Furthermore, this study provides insight into the pathogenesis of hyperuricemia and highlights the anti-hyperuricemic properties of anserine.

## Figures and Tables

**Figure 1 nutrients-15-00969-f001:**
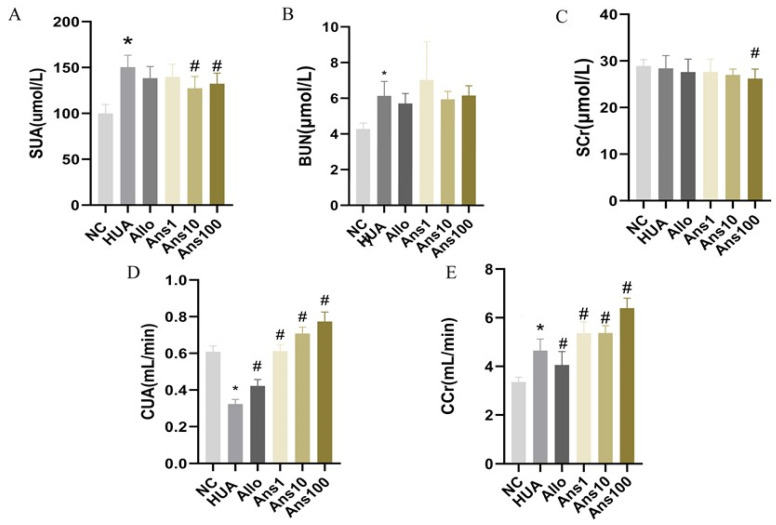
Effect of anserine on hyperuricemia-related indicators. The level of serum (**A**) SUA (serum uric acid); (**B**) BUN (blood urea nitrogen); (**C**) SCr (serum creatinine); (**D**) CUA (uric acid clearance); and (**E**) CCr(creatinine clearance); * *p* < 0.05 indicates that the difference is significant when compared with the NC group. ^#^
*p* < 0.05 indicates that the difference is significant when compared with the HUA group.

**Figure 2 nutrients-15-00969-f002:**
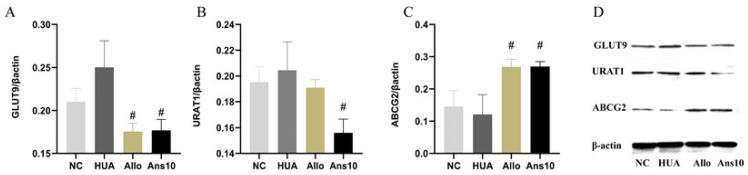
Effect of anserine on kidney urate transport-related proteins. (**A**) GLUT9; (**B**) URAT1; (**C**) ABCG2; and (**D**) Western blot result of GLUT9, URAT1 and ABCG2 protein in each group. ^#^
*p* < 0.05 indicates that the difference is significant when compared to the HUA group.

**Figure 3 nutrients-15-00969-f003:**
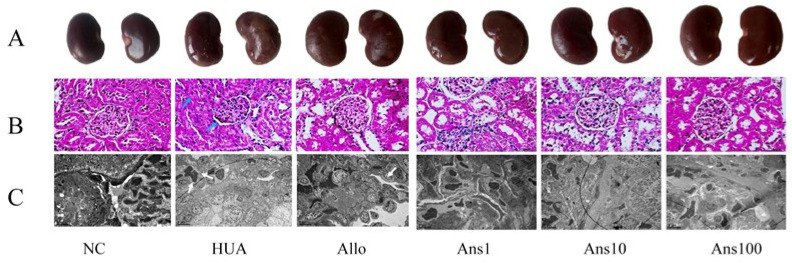
Morphological observation of anserine on kidney pathological damage caused by hyperuricemia. Morphological pictures of different groups of kidneys were observed: (**A**) naked eyes, (**B**) light microscopy and (**C**) electron microscopy.

**Figure 4 nutrients-15-00969-f004:**
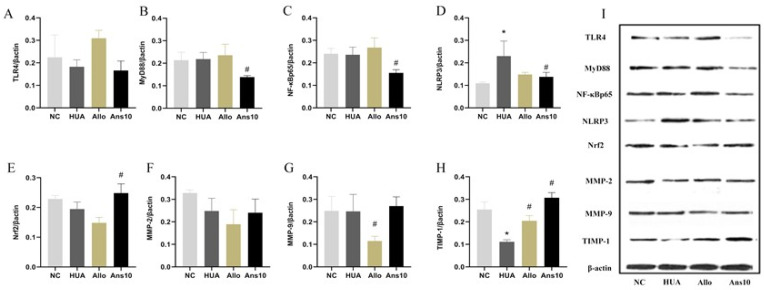
Effect of anserine on the TLR4/MyD88/NF-κB pathway, NLRP3 inflammasome, Nrf2, and MMP-2, MMP-9, and TIMP-1. The relative expression of: (**A**) TLR4; (**B**) MyD88;(**C**) NF-κBp65; (**D**) NLRP3; (**E**) Nrf2; (**F**) MMP-2; (**G**) MMP-9; and (**H**) TIMP-1 among groups. (**I**) Western blot results of TLR4/MyD88/NF-κBp65/NLRP3/Nrf2/MMP-2/MMP-9/TIMP-1 proteins in each group. * *p* < 0.05 indicates that the difference is significant when compared to the NC group. ^#^
*p* < 0.05 indicates that the difference is significant when compared to the HUA group.

**Figure 5 nutrients-15-00969-f005:**
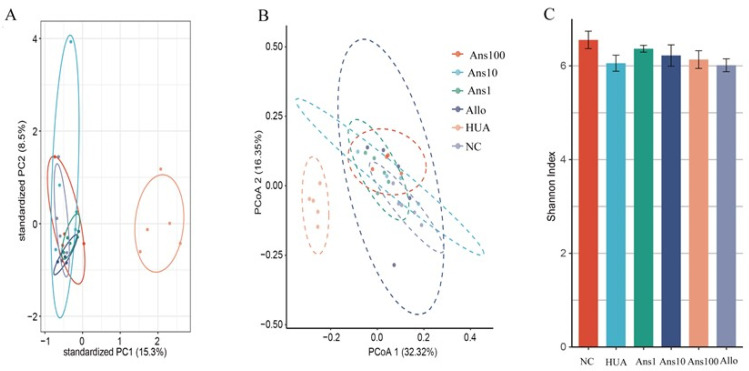
Effect of anserine on the gut microbiota structure and diversity: (**A**) PCA, (**B**) PcoA, and (**C**) Shannon index (*p* < 0.05). (“Ans100” refers to “Anserine 100 mg group”; “Ans10” refers to “Anserine 10 mg group”; “Ans1” refers to “Anserine 1 mg group”; “Allo” refers to “Allopurinol group”; “HUA” refers to “hyperuricemic group” “NC” refers to “Normal control group”).

**Figure 6 nutrients-15-00969-f006:**
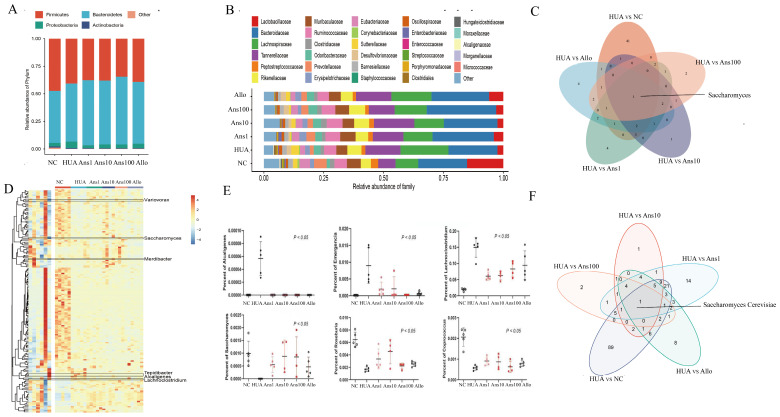
Alterations in the gut microbiota of each group. Abundance distribution: (**A**) phylum level; (**B**) genus level; (**D**) species level; (**E**) The three differential genus with the highest and lowest abundance in the HUA group compared to the Anserine and allopurinol intervention groups. (**C**,**F**) Common genus and species shared with five compared groups (*p* < 0.05).

**Figure 7 nutrients-15-00969-f007:**
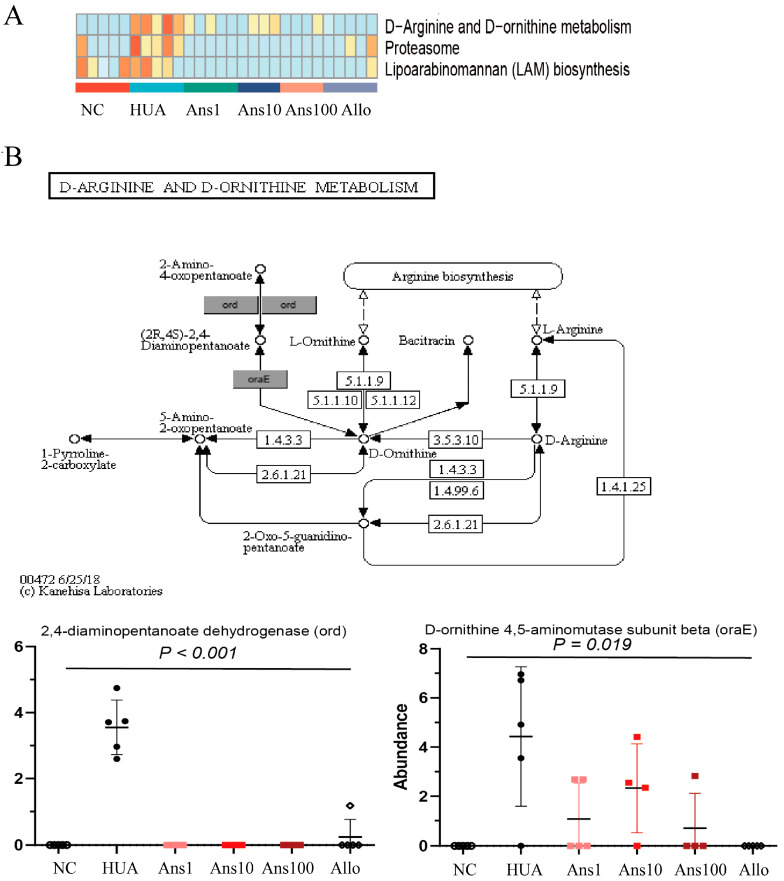
Changes in the metabolic function of gut microbiota: (**A**) three metabolic pathways in the HUA group are richer than other groups, including D-Arginine and D-ornithine metabolism, proteasome and Lipoarabinomannan (LAM) biosynthesis pathways; (**B**) altered genes in the D-arginine and D-ornithine metabolism pathways (*p* < 0.05).

**Figure 8 nutrients-15-00969-f008:**
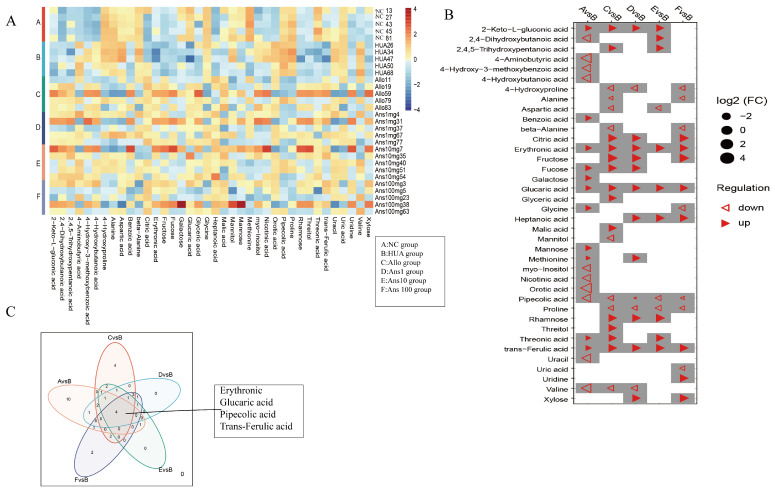
Differential urine metabolites among groups: (**A**) the heatmap of the relative level of 38 differential metabolites in each group; (**B**) the trend changes for 38 differential metabolites in five comparison groups; (**C**) the Venn diagram shows common metabolites for the five comparison groups (*p <* 0.05).

**Figure 9 nutrients-15-00969-f009:**
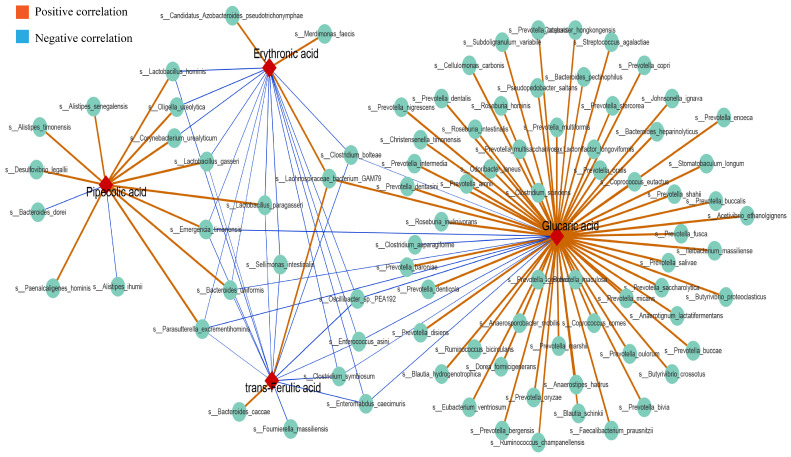
Species that are significantly associated with erythroid acid, glucaric acid, pipecolic acid, and trans-ferulic acid. The orange and blue lines represent positive and negative correlations, respectively (*p* < 0.05).

## Data Availability

The data that supports the findings of this study are available from the corresponding author upon reasonable request.

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
