# Peer review of "Anti-Hyperuricemic Effect of Anserine Based on the Gut–Kidney Axis: Integrated Analysis of Metagenomics and Metabolomics"

_nutrients, 2023, doi:10.3390/nu15040969_

Round 1
Reviewer 1 Report
Manuscript title: Anti-Hyperuricemia Mechanism of Anserine Based on the Gut-Kidney Axis: Integrated Analyses of Metagenomic and Metabolomic
Summary
The purpose of the study was to assess whether the bioactive peptide anserine, which has been demonstrated to improve kidney function and alleviating inflammatory responses through change in the gut microbiota, may reduce hyperuricemia via the gut–kidney axis.
For this aim they established a rat model of hyperuricemia and verifies the ameliorating effect of anserine or allopurinol on hyperuricemia. They also explore the mechanism of anserine-related amelioration of hyperuricemia through gut microbiota in order to assess anserine potential in preventing and treating hyperuricemia.
The topic is interesting to the field, methods are quite correct, language is fine. However, there are some points which need to be addressed or clarified.
Comments
1) Page 2: “When the kidneys are damaged due to hyperuricemia, uric 91 acid and uremic toxins accumulate in the blood.” The sentence is not complete (it should be then instead of when?).
2) Which are the anserine concentrations in fish and at biological model? May the anserine doses used in the study be comparable to life model? The effect of anserine is probably related to its concentration in the model as also demonstrated in this study since different doses show different (more or less significant) effect. How do the Author choose the anserine concentration used in their study? Are there any references about doses?
3) The Author suggest that anserine improves the overall kidney function by decreasing inflammation, oxidative stress and cellular damage in hyperuricemia since anserine decreased the expression of TLR4/MyD88/NF-κB and NLRP3 and elevated Nrf2 and TIMP1 in hyperuricemia rats. Do the Author have any data about oxidative stress or inflammatory factors in the kidney which may confirm the result of activation of these signaling pathways?
4) The Authors evaluated MMP-2 and MMP-9 by western blotting. I have some concerns about this method since the best technique to assess these MMPs is gelatin zymography which can determine whether the MMP is in an active or latent form.
5) The Author reported that the levels of CCr were elevated in the HUA group. How do the Author explain this result?
6) Data about Ans1 and 100 were not reported in Figure 2 and Figure 4. Did they have similar result compared to the middle dose of Anserin?
Author Response
Comments (1): “When the kidneys are damaged due to hyperuricemia, uric acid and uremic toxins accumulate in the blood.” The sentence is not complete (it should be then instead of when?).
Reply: Thank you so much for the suggestion, I revised this sentence as follows “ The uric acid and uremic toxins will accumulate in the blood when kidneys are damaged due to hyperuricemia.”
Comments (2): Which are the anserine concentrations in fish and at biological model? May the anserine doses used in the study be comparable to life model? The effect of anserine is probably related to its concentration in the model as also demonstrated in this study since different doses show different (more or less significant) effect. How do the Author choose the anserine concentration used in their study? Are there any references about doses?
Reply: Thank you so much for these questions, they make me more comprehensive thoughts on my research. It is reported that in general, bonito, bigeye tuna, and southern tuna contain about 1070mg, 1260mg and 636mg of anserine per 100g of fish flesh, respectively[1]. By reviewing the literature, we learned that the doses of anserine used in previous animal experiments ranged from 2mg/kg.bw to 80mg/kg.bw, so we chose the 1mg/10mg/100mg kg·bw as low/medium/high dose of anserine intervention in our experiment. Our results did showed that medium doses of anserine were the most effective in improving hyperuricemia. So we speculate that the effect of anserine in improving hyperuricemia is dose-dependent.
References:
[1] Boldyrev AA, Aldini G, Derave W. Physiology and pathophysiology of carnosine. Physiol Rev. 2013 Oct;93(4):1803-45. doi: 10.1152/physrev.00039.2012. PMID: 24137022.
[2] Chen M, Ji H, Song W, Zhang D, Su W, Liu S. Anserine beneficial effects in hyperuricemic rats by inhibiting XOD, regulating uric acid transporter and repairing hepatorenal injury. Food Funct. 2022 Sep 22;13(18):9434-9442. doi: 10.1039/d2fo01533a. PMID: 35972268.
[3] Taiken Sakano, Ai Saiga Egusa, Yoko Kawauchi, Jiawei Wu, Toshihide Nishimura, Nobuhiro Nakao, Ayumu Kuramoto, Takumi Kawashima, Shigenobu Shiotani, Yukio Okada, Kenichiro Sato, Nobuya Yanai, Pharmacokinetics and tissue distribution of orally administrated imidazole dipeptides in carnosine synthase gene knockout mice, Bioscience, Biotechnology, and Biochemistry, Volume 86, Issue 9, September 2022, Pages 1276-1285, https://doi.org/10.1093/bbb/zbac081
Comments (3): The Author suggest that anserine improves the overall kidney function by decreasing inflammation, oxidative stress and cellular damage in hyperuricemia since anserine decreased the expression of TLR4/MyD88/NF-κB and NLRP3 and elevated Nrf2 and TIMP1 in hyperuricemia rats. Do the Author have any data about oxidative stress or inflammatory factors in the kidney which may confirm the result of activation of these signaling pathways?
Reply: I really appreciate the question you purposed. The TLR4/MyD88/NF-κB pathway and NLRP3 are reported to be the classic proteins related to kidney inflammation caused by hyperuricemia. Nrf2 is a very strong antioxidant in vivo, so I think that detecting the change of Nrf2 expression after anserine intervention can indicate the antioxidant effect of anserine. However, I know It would be nice to have data on inflammation-related factors such as TNF-α Interleukin-1β or oxidative stress-related proteins like SOD and MDA, but I'm sorry we don't have data on that. I will seriously consider this point in my research.
Comments (4): The Authors evaluated MMP-2 and MMP-9 by western blotting. I have some concerns about this method since the best technique to assess these MMPs is gelatin zymography which can determine whether the MMP is in an active or latent form.
Reply: It’s a really good question, thank you so much! Yeah, the gelatin zymography method is an excellent method for the analysis of MMP2 and MMP-9 activity. However, it does not reflect the changes in protein expression. In my research, I want to determine the change of MMP2 and MMP9 expression after anserine intervention in hyperuricemia rats. By observing the changes, we can speculate the ameliorative effect of anserine on the cell injury. Therefore, I suppose western blotting is a better way to achieve that.
Comments (5): The Author reported that the levels of CCr were elevated in the HUA group. How do the Author explain this result?
Reply: Thank you so much for this thought-provoking question. This result is indeed not quite what we expected, but the reasons for this result are understandable. Blood creatinine is a product of muscle metabolism and is normally excreted through the kidneys thus maintaining a homeostasis of creatinine levels in the body. When hyperuricemia occurs, prolonged elevated uric acid can cause urate deposits in the interstitial kidney, which can lead to kidney function damage. This leads to a decrease in the ability of the kidneys to remove creatinine. In our experiments, creatinine clearance, however, showed an increase, and I think it is very likely because when the blood creatinine in hyperuricemic rats starts to rise, the body compensates by increasing the renal creatinine clearance to keep the body's creatinine level at a steady state. So the timing of the measurement also has some influence on the results.
Comments (6): Data about Ans1 and 100 were not reported in Figure 2 and Figure 4. Did they have similar result compared to the middle dose of Anserin?
Reply: Thank you very much for raising this query, this issue really needs to be clarified. As we discussed earlier, based on the changes in the physiological indices associated with hyperuricemia in rats after intervention with anserine, we found that anserine ameliorated hyperuricemia in a dose-dependent manner. Among them, the medium dose of anserine intervention was the most effective, so in order to save time and measurement cost, in the later experiments of related protein expression measurement, macrogenomic sequencing and UPLC-MS measurement, we only selected the samples after in the medium dose of anserine intervention group for further testing. And we think this is also in line with our aim to investigate the mechanism by which anserine improves hyperuricemia.

Reviewer 2 Report
This manuscript describes a remarkably thorough investigation into the role of anserine in hyperuricemia. Each experiment is carefully described, and the results shown clearly in the Figures. In combination, the results demonstrate a clear correlation between anserine levels and anti-hyperuricemic effect. These results should have important implications for human health, so the work will be of broad interest. This work is somewhat outside my area of expertise, but I see no problems with the results, conclusions or structure of the paper. English is of high quality, just a couple minor points listed below.
1. Line 270: similar effects as THOSE of.... " (not THAT). Please check agreements throughout.
2. Line 452: "Emergencia timonensis genus" is all in italics. I don't think the word genus should be italicized here. And similar elsewhere
3. Line 454: Trimethylamine. Should not have capital T. Likewise Ferrulic acid at line 518
Author Response
Comment 1: Line 270: similar effects as THOSE of.... " (not THAT). Please check agreements throughout.
Reply: Thank you so much for the suggestion. I have changed this mistake, for detail, please check line 270.
Comment 2: Line 452: "Emergencia timonensis genus" is all in italics. I don't think the word genus should be italicized here. And similar elsewhere
Reply Thanks a lot for the reminder, I didn't pay attention to this issue. Based on your suggestion, I have changed the italicized formatting of the word labeled genus in the full text, as detailed in the updated version of the manuscript, thank you so much!
Comment3: Line 454: Trimethylamine. Should not have capital T. Likewise Ferrulic acid at line 518
Reply: Thank you very much for the correction. I have already made changes to the corresponding parts according to your suggestions. See details at lines 454,538 and 518 in the updated version of the manuscript.

Reviewer 3 Report
The authors used the rat model to investigate the effect of an anserine-based diet on hyperuricemia. The authors have used various analytical methods to demonstrate that an anserine-based diet has a positive effect on the reversal of hyperuricemia. The work is well characterized.
1) The title requires rework.
2) The English language and grammar throughout the paper require editing and corrections.
Author Response
Comment1: The title requires rework.
Reply: Thank you very much for the offer. After careful consideration, we have changed the title to " Anti-Hyperuricemia Effect of Anserine Based on the Gut-Kidney Axis: Integrated Analyses of Metagenomic and Metabolomic". Please check the detail in the updated version of the manuscript on line 1. Please feel free to talk me if this is more appropriate.
Comment2: The English language and grammar throughout the paper require editing and corrections.
Reply: Thank you very much for your suggestions. Some grammatical description errors in the article have been fixed.

Round 2
Reviewer 1 Report
Manuscript title: Anti-Hyperuricemia Mechanism of Anserine Based on the Gut-2 Kidney Axis: Integrated Analyses of Metagenomic and Metabo-3 lomic
In the revised version there are still some points which need to be addressed or clarified.
The Authors answered the erased questions but they did not include changes in the manuscript. For example, I would suggest to add comment in the text relating to the selection of the anserine concentration and the references about doses (question 2: How do the Author choose the anserine concentration used in their study? Are there any references about doses?).
Similarly, I would suggest to add comment in the manuscript explaining the increased levels of CCr in the HUA group (question 5: The Author reported that the levels of CCr were elevated in the HUA group. How do the Author explain this result?).
The title “Anti-Hyperuricemia Effect of Anserine…” should be changed in “Anti-Hyperuricemic Effect of Anserine…”

Author Response
Thank you very much for such constructive suggestions.I have made changes or explanations based on your suggestions at the appropriate places in the newly uploaded version of the manuscript, please check.